# Mechanism of the Effect of High-Intensity Training on Urinary Metabolism in Female Water Polo Players Based on UHPLC-MS Non-Targeted Metabolomics Technique

**DOI:** 10.3390/healthcare9040381

**Published:** 2021-04-01

**Authors:** Lei-lei Wang, An-ping Chen, Jian-ying Li, Zhuo Sun, Shi-liang Yan, Kai-yuan Xu

**Affiliations:** 1College of Physicial Education, Shanxi University, Taiyuan 030006, China; wll728ca@163.com (L.-l.W.); yanshiliangyy@163.com (S.-l.Y.); xky13603549207@126.com (K.-y.X.); 2Department of health and Natural Sciences, Gdansk University of Physical Education and Sport, 80-336 Gdańsk, Poland; zhuo.sun@awf.gda.pl

**Keywords:** high-intensity training, LC-MS, urine, water polo

## Abstract

Objective: To study the changes in urine metabolism in female water polo players before and after high-intensity training by using ultra-high performance liquid chromatography-mass spectrometry, and to explore the biometabolic characteristics of urine after training and competition. Methods: Twelve young female water polo players (except goalkeepers) from Shanxi Province were selected. A 4-week formal training was started after 1 week of acclimatization according to experimental requirements. Urine samples (5 mL) were collected before formal training, early morning after 4 weeks of training, and immediately after 4 weeks of training matches, and labeled as T1, T2, and T3, respectively. The samples were tested by LC-MS after pre-treatment. XCMS, SIMCA-P 14.1, and SPSS16.0 were used to process the data and identify differential metabolites. Results: On comparing the immediate post-competition period with the pre-training period (T3 vs. T1), 24 differential metabolites involved in 16 metabolic pathways were identified, among which niacin and niacinamide metabolism and purine metabolism were potential post-competition urinary metabolic pathways in the untrained state of the athletes. On comparing the immediate post-competition period with the post-training period (T3 vs. T2), 10 metabolites involved in three metabolic pathways were identified, among which niacin and niacinamide metabolism was a potential target urinary metabolic pathway for the athletes after training. Niacinamide, 1-methylnicotinamide, 2-pyridone, L-Gln, AMP, and Hx were involved in two metabolic pathways before and after the training. Conclusion: Differential changes in urine after water polo games are due to changes in the metabolic pathways of niacin and niacinamide.

## 1. Introduction

Since its first introduction by Nicholson [1] in 1999, metabolomics has gradually developed and gained significance as a tool in systems biology. It can respond more directly to the physiological status of organisms and changes in the types and quantities of metabolites with molecular weights of 1000 Da or less. Mass spectrometry (MS) is used in metabolomics detection because it can detect multiple metabolites in a single experiment, given its high sensitivity and large dynamic range of monitoring. It can detect and track metabolite changes correlated with the target state in a global, non-targeted analysis [2,3,4,5].

Among many other applications, metabolomics is applied in sports, for monitoring exercise intensity and body metabolism [6], performance prediction [7,8], exercise in disease diagnosis and treatment [9], and sports nutrition supplementation [10]. In China, metabolomics is now being applied in choosing exercise and nutritional supplementation [11], athlete selection [12], and particularly in exploring the metabolic mechanisms of athletes [13,14] and the improvement in diseases by exercise [15,16].

Water polo is a physical sport combining swimming, handball, and volleyball. It is a type of metabolic energy supply involving the phosphate source system, lactic energy system, and aerobic oxidation system [17]. A human’s athletic ability largely depends on the body’s metabolism. Different exercise routines have differential changes in the metabolism. The changes in endogenous metabolites facilitating body transformation during exercise also have different effects on the metabolism, depending upon the type of exercise routine. Little has been reported about the effects of water polo training on the metabolic mechanisms of small molecules in the body This study analyzes the metabolic effects of water polo by identifying the changes in urinary metabolites in players before and after training, based on LC-MS technology. It also provides theoretical references for developing water polo training programs and improving sports training efficiency.

## 2. Materials and Methods

### Subjects and Grouping

Twelve female youth water polo players (excluding two goalkeepers) from Shanxi Province, with similar training levels and years of training, were selected. They were in good health, with no metabolic diseases. They did not consume any drugs, smoke, alcohol, stimulant or functional drinks, and slept early starting from 1 month before the experiment until its completion. To reduce experimental variability, all athletes were trained at the Shanxi Sports Center throughout the experiment, and their team teachers were responsible for their diet (The daily diet we provided the athletes included: eggs, milk, beef, pork chops, mutton, sea cucumbers, Doufu, broccoli, pumpkin, bone soup, an apple or orange, and rice or noodles) and living. The basic information of the athletes is shown in Table 1.

## 3. Training Program

To ensure objectivity and minimize error, the experiment was designed to include adaptive and formal training for 5 weeks. All athletes strictly followed the experimental requirements for 1 week of adaptive training before entering the formal training phase. Four complete water polo training weeks were selected for the formal experiment, and the training intensity and time were the same as normal training.

Training program: Monday to Saturday: training; Sunday: rest, for a total of 4 weeks. Morning: 30-min wall patting and warm-up training. Comprehensive strength training on land: bench press, squat, pull-up, static lumbar, 5 kg barbell stepping, frog jump, touch ground vertical jump, and comprehensive barbell exercises with stretching and relaxation. Water training: warm-up: 200 m mixed swimming, freestyle kicking, and stroke technique; main training: 10 × 200 m freestyle, four sets of 3 × 400 m freestyle, four sets of step swimming (400-200-100-50), 25 m freestyle sprint; comprehensive water pull training and water relaxation. Afternoon: basic water polo training (freestyle kicking + single and double alternating stirrups, leg clipping, cross-sealing hands, 25 m shift pick and jump break, emergency stop start, freestyle directional swimming, fast frequency freestyle hands combined with single stirrups), tactical exercises (counterattacking more and less, big field round trip, six to five, five defense six, position attack running, crossover, cut-in practice), passing and receiving (front and backhand, six to five), position attack shooting, water relaxation, and land pulling relaxation. Warm-ups and intra-group games were held on Saturday.

The swimming and diving pool is an indoor artificial pool; the water quality indicators of the pool meet the management standards of the pool (GB37487, 2019/11): residual chlorine 0.3–1.0 mg/L, pH 7.0–7.8, combined chlorine ≤0.4 mg/L, total number of colonies ≤200, turbidity ≤1, cyanouric acid ≤50 mg/L, carbamide ≤3.5 mg/L [18].

### 3.1. Experimental Grouping and Sampling

We collected 5 mL of urine in 10 mL EP tubes on the first day of training, last day of training, and in the immediate post-competition period after 4 weeks of training, and labeled them as T1, T2, and T3, respectively. The samples were first stored in liquid nitrogen and then transferred to a −80 °C refrigerator (BIOBASE, Jinan, China).

After thawing at 4 °C, an appropriate amount of the sample was added to pre-chilled methanol/acetonitrile/water solution (2: 2:1, *v*/*v*), vortexed and mixed, sonicated at a low temperature for 30 min, left at −20 °C for 10 min, and centrifuged (5430R, Eppendorf, Germany) at 14,000× *g* for 20 min at 4 °C. The supernatant was vacuum dried, 100 μL of aqueous acetonitrile solution (acetonitrile:water = 1:1, *v*/*v*) was added for mass spectrometry analysis, vortexed, and centrifuged at 14,000× *g* for 15 min at 4 °C. The supernatant was collected for analysis.

### 3.2. Detection Method

#### Reagents and Instrument

AB Triple TOF 6600 mass spectrometer (SCIEX, USA); Agilent 1290 Infinity LC ultra-high pressure liquid chromatograph (Agilent Technologies, USA); low-temperature high-speed centrifuge (5430R, Eppendorf, Germany); chromatographic column: Waters, ACQUITY UPLC BEH Amide 1.7 μm (Waters, USA, 2.1 mm × 100 mm column (Waters, USA); acetonitrile (Merck, CAN); ammonium acetate (Sigma-Aldrich, Shanghai, China).

### 3.3. Chromatography-Mass Spectrometry Analysis

#### 3.3.1. Chromatographic Conditions

The samples were separated on an Agilent 1290 Infinity LC ultra-high performance liquid chromatography system (UHPLC) HILIC column (Agilent Technologies, USA); column temperature 25 °C; flow rate 0.5 mL/min; injection volume 2 μL; mobile phase composition A: water + 25 mM ammonium acetate (Sigma-Aldrich, Shanghai, China); +25 mM ammonia (Sigma-Aldrich, Shanghai, China); B: acetonitrile; gradient elution procedure: 0–0.5 min, 95% B; 0.5–7 min, B changed linearly from 95% to 65%; 7–8 min, B changed linearly from 65% to 40%; 8–9 min, B remained at 40%; 9–9.1 min, B changed linearly from 40% to 95%; 9.1–12 min, B remained at 95%. The samples were placed in a 4 °C autosampler throughout the analysis. To avoid the effects caused by fluctuations in the instrument’s detection signal, a random order was used for the continuous analysis of samples. QC samples were inserted in the sample queue for evaluating the stability of the system and the reliability of experimental data [19,20,21].

#### 3.3.2. Q-TOF Mass Spectrometry Conditions

The AB Triple TOF 6600 mass spectrometer was used to obtain the primary and secondary spectra of the samples. The ESI source conditions after HILIC chromatographic separation were as follows: Ion Source Gas 1 (Gas1): 60, Ion Source Gas 2 (Gas2): 60, Curtain gas (CUR): 30, source temperature: 600 °C, IonSapary Voltage Floating (ISVF) ± 5500 V (positive and negative modes); TOF MS scan m/z range: 60–1000 Da, product ion scan m/z range: 25–1000 Da, TOF MS scan accumulation time 0.20 s/spectra, product ion scan accumulation time 0.05 s/spectra. The secondary mass spectra were acquired using information-dependent acquisition (IDA) and adopted high sensitivity mode; Declustering potential (DP): ±60 V (positive and negative modes); Collision Energy: 35 ± 15 eV, IDA settings: Exclude isotopes within 4 Da; and Candidate ions to monitor per cycle: 10.

#### 3.3.3. LC-MS Data Analysis

The raw data in Wiff format were converted to mzXML format by ProteoWizard (SOUREFORGE, San Diego, CA, USA). XCMS software (Scripps Research Institute, San Diego, CA, USA) was used to perform peak alignment, retention time correction, and peak area extraction. The data extracted by XCMS were checked for completeness; metabolites with more than 50% missing values within the group were removed from subsequent analysis, extreme values were removed, and the data were normalized for total peak area to ensure parallelism for comparison between samples and metabolites. The preprocessed data were imported into SIMCA-P 14.1 (Umetrics, Umea, Sweden) for pattern recognition, and the data were preprocessed by Pareto-scaling and subjected to multidimensional statistical analysis, including unsupervised Principal Component Analysis (PCA), supervised partial least squares discriminant analysis (PLS-DA) and orthogonal partial least squares discriminant analysis (OPLS-DA). Between-group differential metabolite screening was performed with OPLS-DA VIP > 1 and *p* < 0.05 as significant differential metabolite screening criteria.

All data were expressed as mean ± variance (X ± SD). The *t*-test was performed using SPSS 21.0 (SPSS Inc., Chicago, IL, USA) to analyze the differences between groups.

## 4. Results

### 4.1. Stability Evaluation of Subjects’ Urine LC-MS Assay System

The results of the total ion chromatogram (TIC) spectral overlap (A) and the correlation spectrum of QC samples (B) are shown in Figure 1. The response intensity and retention time of the peaks of TIC overlapped, and the correlation coefficients of QC samples were greater than 0.9, indicating good correlation. This suggested that the instrument analysis system of this test was stable, with low error, good repeatability, and stable and reliable test data.

### 4.2. Multivariate Statistical Analysis

Partial least squares discriminant analysis (PLS-DA) is a supervised statistical method of discriminant analysis that amplifies the differences between groups and filters out differential lipids associated with grouping from the data set. Figure 2A,D shows the PLS-DA model validation plot for this experiment. Model validation determines whether the analyses of PLS-DA and OPLS-DA results are meaningful. In this study, 200 response-ranking tests were used for the model. When the Q2 and R2 values are closer to 1, the model has higher validity. The parameters of this study for this model validation, shown in Table 2, indicate that the model is robust.

Orthogonal partial least squares discriminant analysis (OPLS-DA) was subsequently performed. OPLS-DA, a modified analysis method for PLS-DA, can eliminate metabolic changes caused by factors unrelated to the experiment to improve the resolution and validity of the model. There are two principal components on the OPLS-DA score plot, i.e., projected principal component and orthogonal principal component. OPLS-DA maximizes the between-group variation on t [1] so that the between-group variation can be directly distinguished from t [1], while the within-group variation is reflected on the orthogonal principal component t [1]. Figure 2 shows model validation plots, OPLS-DA score plots, and metabolite clustering plots for the two groups comparing pre-training to post-competition, and post-training to post-competition in terms of urine metabolism. The OPLS-DA plot in Figure 2 clearly shows that the urine samples can be differentiated into two groups before and immediately after water polo training (Figure 2B), as well as after training and immediately after competition (Figure 2E). This accords with the response of the cluster analysis: Most urine sample metabolites can be closely clustered at the same moment and differentiated from the urine sample metabolites at another moment, indicating that water polo competition impacts urine metabolism.

### 4.3. Screening of Differential Metabolites in Urine by LC-MS

With OPLS-DA VIP > 1 and *p* < 0.05 as significantly different values, metabolites between different groups were screened using online databases such as HMDB (http://www.hmdb.ca/) (accessed on 1 September 2020) and KEGG (https://www.kegg.jp) (accessed on 1 September 2020).

Table 3 shows the results and trends of metabolite screening. Figure 3 shows the FC diagram of metabolite changes among groups. After screening and statistical analysis, 27 differential metabolites were screened in this study (Table 3). A total of 24 metabolites were screened before and after training, and 10 differential metabolites were screened after 4 weeks of training compared with the immediate post-competition period, among which there were seven co-differential metabolites, namely N-acetylglutamine, decanoyl-L-carnitine, TMAO, Hx, NAM, N-acetyl-L-histidine, and N6-methyl-adenosine. The reduction in the number and type of urinary metabolites after training indicated that 4 weeks of high-intensity water polo training produced a significant intervention effect on the body’s urinary metabolism and that the body’s metabolic changes were more stable compared to the pre-training period.

### 4.4. Analysis of Differential Metabolites and Screening of Potential Biomarkers in Urine after Water Polo Games

#### 4.4.1. Correlation Analysis of Differential Metabolites

Correlation analysis facilitates the measurement of metabolic proximities between significantly different metabolites (VIP > 1, *p* < 0.05) and helps in understanding the mutual regulation between metabolites during the change of biological state. Figure 4 shows the analysis of the correlation results of 27 significant differential metabolites in urine after the water polo game. Red color indicates positive correlation, and blue color indicates negative correlation. The darker color is proportional to the absolute value of correlation coefficient and indicates a stronger correlation. It suggests that the up-regulation or down-regulation of endogenous metabolites in human urine after the water polo game causes the related metabolites to increase or decrease.

#### 4.4.2. Hierarchical Clustering of Differential Metabolites

Figure 5 shows hierarchical clustering plots of significant differential metabolites in urine at T1 vs. T3 (A) and T2 vs. T3 (B) moments. It offers a visual depiction of the trend of metabolic changes in human urine at different intervals of water polo competition. The horizontal coordinate indicates the sample and the vertical coordinate the intensity of metabolite expression; red indicates positive correlation, blue indicates negative correlation, and the darker color indicates higher metabolite intensity. To observe the metabolite change trends of each group at different moments more clearly, the mean metabolite expression clustering plots (C and D) among groups were obtained on the basis of independent sample clustering plots (A and B). The average relative content expression between groups (right) indicates that the differential metabolites before and after training can be clustered into two major groups with those after the game, showing completely opposite expression patterns. This indicates that water polo has a significant effect on human urine; it undermines the steady state of the body, leading to changes in urinary metabolic pathways.

### 4.5. Analysis of the Metabolic Pathway of Urinary Differentials by Water Polo Exercise

MetaboAnalyst 5.0 (http://www.metaboanalyst.ca, accessed on 1 September 2020) was used to perform metabolic pathway enrichment analysis of differential urinary metabolites. The above-obtained differential metabolites were imported into Pathway Analysis and metabolic pathway maps were obtained. The horizontal coordinate pathway impact indicates the importance value of metabolic pathways obtained from topological analysis, and the vertical coordinate, logP, indicates the significance level of metabolic pathway enrichment analysis. In this study, Pathway Impact > 0.05 combined with *p* < 0.05 was used as the criterion for metabolic pathway analysis.

#### 4.5.1. Metabolic Pathway Analysis of Urinary Differential Metabolites between Pre-Training and Immediate Post-Competition

A total of 24 metabolites were involved in 16 metabolic pathways before training compared to the immediate post-competition period (Table 4). In combination with the above screening conditions, two metabolic pathways were found to be involved in water polo post-competition metabolism under non-training conditions (Figure 6). Among the metabolic differentials, NAM, 1-methylnicotinamide, and 2-pyridone were involved in the metabolic pathways of nicotinate and nicotinamide metabolism, and L-Gln, AMP, and Hx were involved in purine metabolism.

#### 4.5.2. Metabolic Pathway Analysis of Urinary Differential Metabolites in the Immediate Post-Training and Post-Competition Periods

Three metabolic pathways involving 10 differential metabolites were observed after 4 weeks of training compared to the immediate post-competition period (Table 5). Only one metabolic pathway met the screening criteria (Figure 7), namely nicotinate and nicotinamide metabolism, in which the differential metabolite nicotinamide was involved.

## 5. Discussion

Urine is ideal for metabolomics studies because of its chemical complexity, high number of metabolites, high concentration, and low influence by proteins or lipids [22]. For human studies, urine has the advantage of being collected in large quantities, noninvasively and continuously over a period to provide more complete information than blood [23]. Metabolomics is not yet widely used in sports. Compared to traditional bioassay techniques, metabolomics requires a smaller sample size, analyzes a larger number of metabolites, and provides a broader view of metabolic characteristics. In sports science research and sports training, the use of non-targeted metabolomics technique helps observe metabolite fluctuations related to energy utilization pathways. Understanding these changes and identifying new biomarkers enable us to gain insight into collective adaptive metabolic regulation and provide a basis for future improvements in exercise programs to improve athletic performance. With this theoretical basis, this study chose non-targeted metabolomics to identify and analyze urinary metabolites in humans immediately after water polo matches in comparison with those before and after 4 weeks of training. The biological characteristics and metabolic mechanisms of water polo exercise on urinary differential metabolites in trained and untrained states were deciphered to realize exercise training monitoring, adjust training methods and loads, and assess the actual water polo training effect, thus reducing the risk of sports training.

### 5.1. Pathway Analysis of Purine Metabolism

Urinary L-Gln, Hx, and adenosine monophosphate levels were significantly upregulated after water polo games compared to pre-training.

L-Gln is the most abundant and versatile amino acid in the body and is important for intermediate metabolism, inter-organ nitrogen exchange for inter-tissue ammonia (NH3) transport, and pH homeostasis. Gln is a substrate in nucleotide synthesis (purine, pyrimidine), nicotinamide adenine dinucleotide (NADPH), and antioxidants, to maintain cellular tissue integrity and biosynthesis [24]. Approximately 80% of the body’s Gln is stored in skeletal muscles at a concentration 30 times that of the plasma Gln content [25]. Thus, skeletal muscle is an important tissue for the storage, synthesis, and release of Gln. Gln is considered the “fuel of the immune system” [26], and the significant decrease in Gln levels in exercising organisms may be related to the disruption of immune metabolism caused by overtraining, while oral Gln may promote the recovery of immune function after exercise [27]. The effects of exercise mode, duration, and intensity on the degree of changes in Gln metabolism in the organism also vary. Plasma Gln levels decrease significantly during or after prolonged endurance exercise (e.g., marathon) [28]. One study found a significant decrease in urinary Gln levels in rats after incremental loading to fatigue exercise (3 h) [29]. In contrast to endurance exercise, plasma Gln levels in humans increased after short-term (< 1 h) high-intensity exercise. Eriksson reported an increase in plasma Gln levels from 538 to 666 μmol/L in subjects after 45 min of incremental load cycling exercise [30]. Plasma Gln levels were lower in overtrained athletes compared to healthy trained athletes [31]. Six weeks of progressive incremental training significantly improved plasma Gln levels [32], and the increase in Gln levels may be related to the training of muscle contractions with increased amounts of intermediates in the TCA cycle [33], which leads to higher Gln synthesis and release. The above findings are consistent with those in the present study. The formal water polo game has four sessions, 8 min each, with a game time of 32 min [34], and the whole process includes various intensities of swimming, treading, passing, catching, and shooting, which is a high-intensity, intermittent exercise pattern. In this study, urine Gln levels immediately after the water polo game were significantly higher compared to before training. This result indicated that this training routine was reasonable and did not cause excessive immune damage to the athletes’ body. In fact, 4 weeks of regular training improved the efficiency of energy utilization to some extent.

Hx is a natural purine derivative and a component of nucleic acids in the form of inosine in the anticodon of tRNAs. Most studies on Hx are based on serum or urine analysis. Hx is directly related to the amount of adenosine triphosphate (ATP) consumed intracellularly and thus is a good biomarker for the evaluation of muscle fatigue [35]. With the hydrolysis of ATP during high-intensity exercise, ADP inevitably increases and the body is hypoxic, causing further metabolism of ADP to produce Hx. However, when the body’s demand for energy is reduced and oxygen supply is sufficient, ATP can be synthesized rapidly and ADP accumulation is reduced, which in turn decreases Hx levels [36,37]. Moreover, as the duration and intensity of exercise increases, the deamination of adenosine monophosphate (AMP) is enhanced, and the accumulation of the deamination products inosine monophosphate (IMP) and NH3 and the concentration of purine metabolism increases, leading to fatigue. AMP deaminase deficiency makes the body more susceptible to fatigue, suggesting a key role of AMP deaminase in the fight against fatigue [38]. In addition, reduced ATP flux is a major feature of metabolic damage after severe hypoxia or ischemia in the body, and Hx, as a major extracellular metabolite, can be used to monitor changes in ATP concentration [39]. A study by SAHLIN showed that after eight men pedaled a power bike at 44% and 72% of maximum oxygen uptake for 6 min until they reached exercise fatigue at 98% of maximum oxygen uptake, their plasma Hx levels were significantly increased [40]. Harkness [41] used high performance liquid chromatography (HPLC) to find a 20-fold increase in urinary Hx levels after exercise compared to pre-exercise. The relative levels of IMP in urine immediately after this water polo match were significantly upregulated compared to pre-training, and the relative levels of Hx were also upregulated to 6.73 times that of the pre-training levels (FC = 6.73). This indicates that as the water polo game continues, the body’s ATP metabolism is enhanced, and the body’s hypoxic state leads to a lower rate of ATP synthesis and the cumulative metabolism of ADP generates a large amount of Hx, triggering exercise fatigue. The post-training urine Hx level in this study was 3.47 times higher than that in the immediate post-competition period (FC = 3.47). Although the relative level of immediate post-competition Hx was still upregulated after 4 weeks of training, the degree of upregulation was reduced compared to the pre-training level, indicating that the anaerobic metabolic capacity and fatigue resistance of the athletes gradually increased after 4 weeks of training. Therefore, Hx can be used as a direct indicator for assessing sports fatigue in water polo games, and fatigue recovery of athletes should be highlighted in the post-competition period.

### 5.2. Analysis of Niacin and Nicotinamide Metabolism

There were 24 urinary differential metabolites after pre-training water polo exercise, six of which were involved in two metabolic pathways. NAM, N1-methylnicotinamide, and 2-pyridone were involved in niacin and nicotinamide metabolism; L-Gln, AMP, and Hx were involved in purine metabolism.

The relative levels of NAM and N1-methylnicotinamide decreased significantly and 2-pyridone increased significantly after water polo games in untrained conditions. NAM and niacin (NA) are both water-soluble derivatives of vitamin B and pyrimidine. NAM exists in the blood as NA amide compounds. Both niacinamide and niacin are derived from food; niacinamide from animal diets and niacin from plant diets. The kidneys can directly convert NAM to NADPH+ to participate in respiration, while the excess NA in the organism is methylated and eventually excreted in urine as N1-methylnicotinamide and 2-pyridone. NA and NAM are components of coenzyme I (NAD+) and coenzyme II (NADP+) in the human body, which act as electron carriers or hydrogen transporters in biological redox and are involved in lipid and sugar metabolism, energy metabolism, oxidative stress, and inflammation regulation [42,43]. Exogenous supplementation of NAM can improve NAD+ utilization, delay skeletal muscle aging, and facilitate cellular metabolism [44]. Increasing NAD+ levels promotes sirtuin 1 (SIRT1) activity, leading to deacetylation, which activates the peroxisome proliferator-activated receptor γ-coactivator 1-α (PGC-1α) [45]. SIRT1 controls lipid and energy metabolism, acts as a negative regulator of triglyceride (TG) synthesis [46], and stimulates fatty acid (FA) oxidation [47]. The beneficial effects of NR and NAM on blood TG and FFA levels can be explained by elevated NAD+ levels because NAD+ promotes SIRT1 activity and mitochondrial biogenesis, thereby enhancing mitochondrial FA oxidation and decreasing TG synthesis. On the other hand, water polo has a high-intensity, intermittent activity pattern and its exercise pattern and duration are great challenges for the aerobic and anaerobic metabolic pathways of the organism of athletes during competition [48]. During long-distance swimming competitions (15–22 min), the athlete relies mainly on aerobic metabolism of sugar (70%) and glycolysis for energy supply (20%), and aerobic lipid metabolism for energy supply is 8% [49,50]. Therefore, NAM not only increases the body’s supply of energy by participating in lipid metabolism, but also acts as a substrate for sugar supply to recharge the exercising organism. In the present study, urinary NAM levels were significantly lower in the immediate post-competition period compared to the pre-training period and post-training period. It was hypothesized that with the continuation of exercise, the glycogen metabolism of athletes was exhausted and NAM was called upon as a metabolic substrate to supply energy to the body. With the decrease in blood glucose levels, glucagon secretion in large amounts led to an increase in FFA levels, and part of NAM was consumed in energy metabolism to enhance FA oxidative energy supply, resulting in a decrease in the relative content of NAM before and after training. This result agrees with the characteristics of water polo games—high intensity and long duration.

Higher levels of nicotinamide-like metabolites have been found in outstanding athletes [51]. Urinary levels of methylnicotinamide were significantly higher in swimmers who reached the finals than in non-finalists [52]. Urinary metabolites identified during the immediate post-competition period in this study had significantly lower levels of the metabolite N-methylnicotinamide compared to those screened in the morning urine before training. The reason for this could be that N-methylnicotinamide is both a downstream metabolite of NAM and an upstream substance of 2-pyridone. Thus, it acts as an energy substrate to supply energy to the exercising athlete and is further metabolized to produce 2-pyridone, thereby leading to a significant downregulation of N-methylnicotinamide after exercise. In contrast, after 4 weeks of regular high-intensity exercise training, no significant changes in N-methylnicotinamide were screened, and the types of urinary differential metabolites were significantly reduced, indicating that the internal environment of the exercising athlete gradually tends to homeostasis with the continuation of exercise training. On the other hand, it can also indirectly indicate that athletes’ exercise capacity has improved after 4 weeks of regular exercise training. Related studies suggest that niacin and nicotinamide metabolism also deserve attention as potential indicators for metabolic studies in high-intensity, endurance exercise [52,53].

Meanwhile, some studies have shown that the conversion of NAM into N-methylnicotinamide and metabolites such as 2-pyridone and 4-pyridone and their excretion in urine is a manifestation of too much NAM or NA in the diet, and long-term consumption of such foods in healthy individuals has side effects on the body [54], such as diabetes [55] and liver damage [56]. Meanwhile, Burke [57] emphasized that carbohydrate intake for water polo athletes should be treated critically, and the daily carbohydrate intake should be expressed in grams per kilogram body weight of the athlete to adapt to the different degrees of fitness of the athlete. Costill [58] stated that swimmers consuming lower carbohydrates cannot sustain high intensity training. Higher carbohydrate dietary intake is also supposed to facilitate athletic training and may improve exercise levels by increasing muscle glycogen reserves. In this study, 2-pyridone levels were significantly higher after 4 weeks of training and dietary intake compared to pre-training. However, no changes in 2-pyridone metabolism were observed when comparing post-training to immediate post-competition period. This theoretical result may be related to the higher protein, dairy, and meat content of the athletes’ nutritional meals. We confirmed the same with the team dietary director after this study. Therefore, this study recommends that athlete meals be adjusted to a reasonable carbohydrate to protein ratio [59,60].

## 6. Conclusions

Based on LC-MS metabolomic assays and multiple analysis methods, 27 differential metabolites and two metabolic pathways were screened after water polo matches compared to before and after training. The 4-week regular training chosen for this experiment was scientific and reasonable, which improved the fatigue resistance and exercise capacity of the players. The differential changes in urine after matches were related to the changes of the common metabolic pathway before and after training: niacin and niacinamide metabolism.

## Figures and Tables

**Figure 1 healthcare-09-00381-f001:**
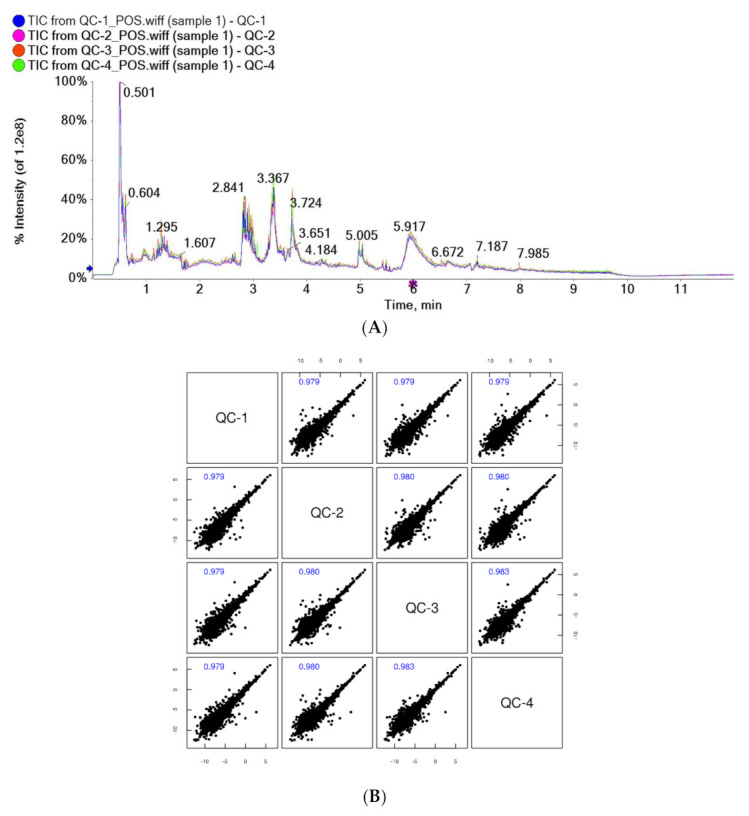
Overlapping spectra of TIC of QC samples in positive ion mode (**A**); correlation spectra of QC samples in positive ion mode (**B**).

**Figure 2 healthcare-09-00381-f002:**
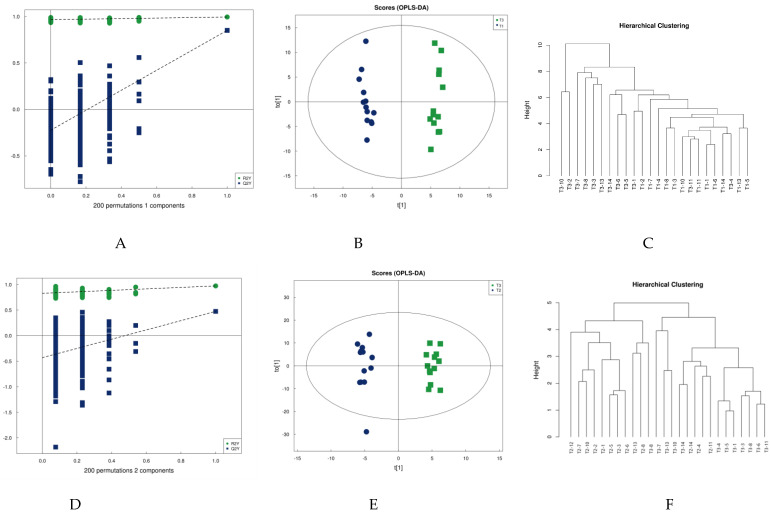
PLA-DA permutation test (**A**,**D**), OPLS-DA scores (**B**,**E**), and cluster analysis (**C**,**F**).

**Figure 3 healthcare-09-00381-f003:**
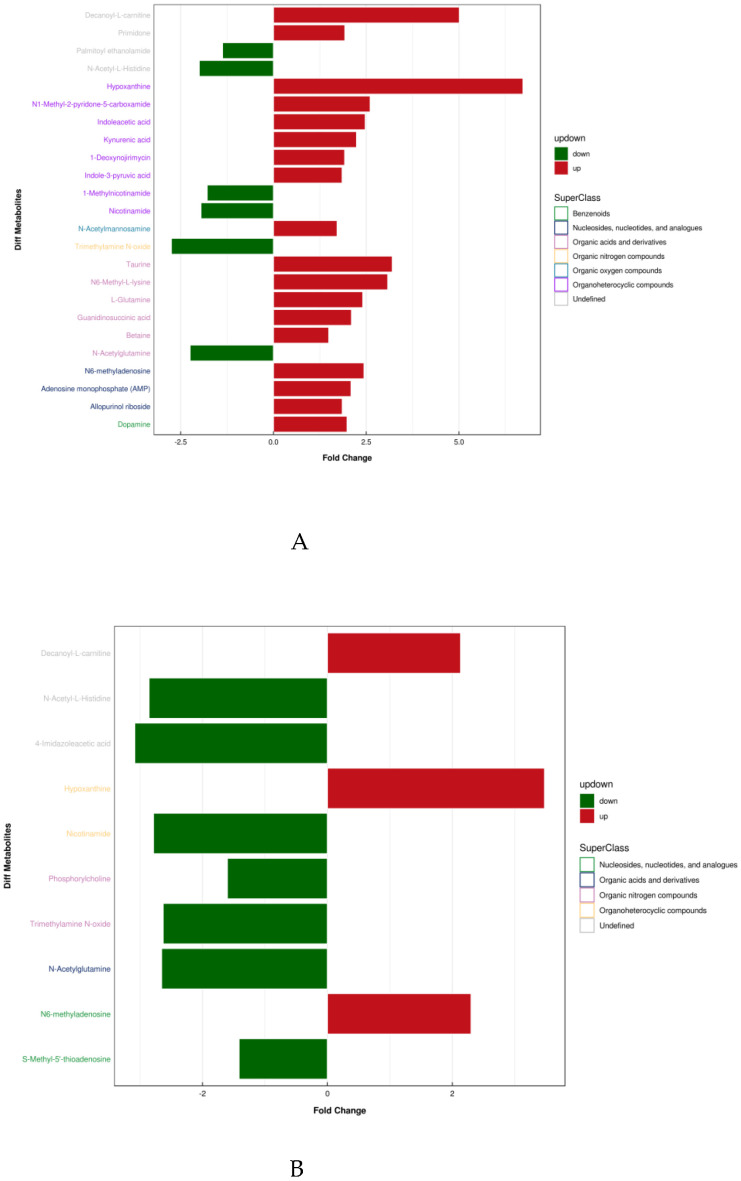
Differential metabolite super classification and FC diagram. (A: T2 vs T1, B: T3 vs T1).

**Figure 4 healthcare-09-00381-f004:**
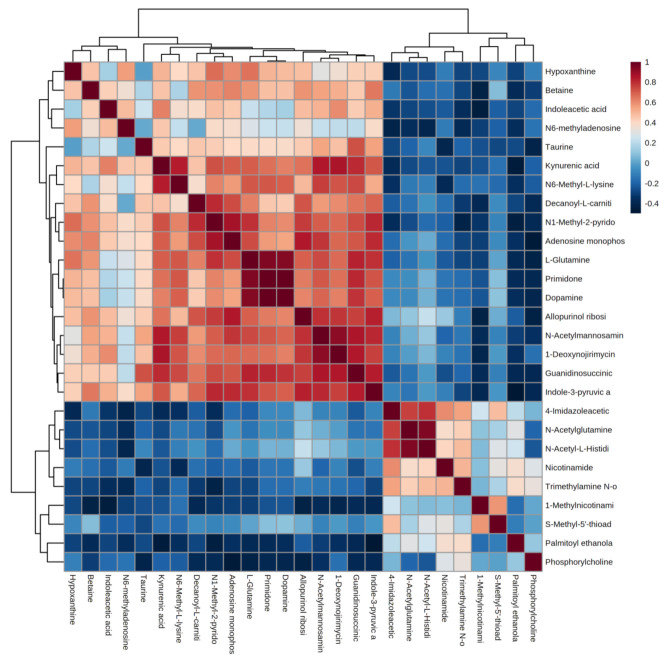
Correlation analysis of differential metabolites after water polo games.

**Figure 5 healthcare-09-00381-f005:**
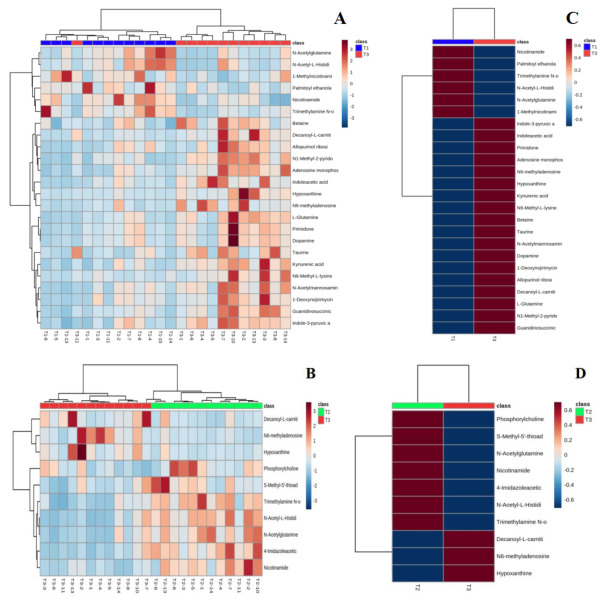
Urine metabolite clustering plots (**A**,**B**) and mean metabolic expression clustering plots between groups (**C**,**D**).

**Figure 6 healthcare-09-00381-f006:**
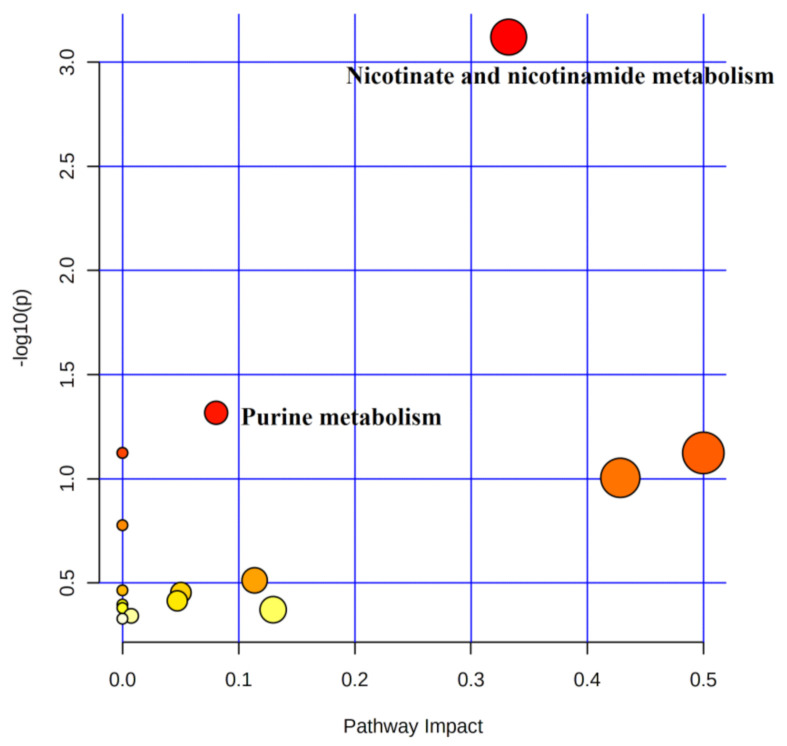
Metabolic pathways of urinary differentials before training compared to immediate post-competition.

**Figure 7 healthcare-09-00381-f007:**
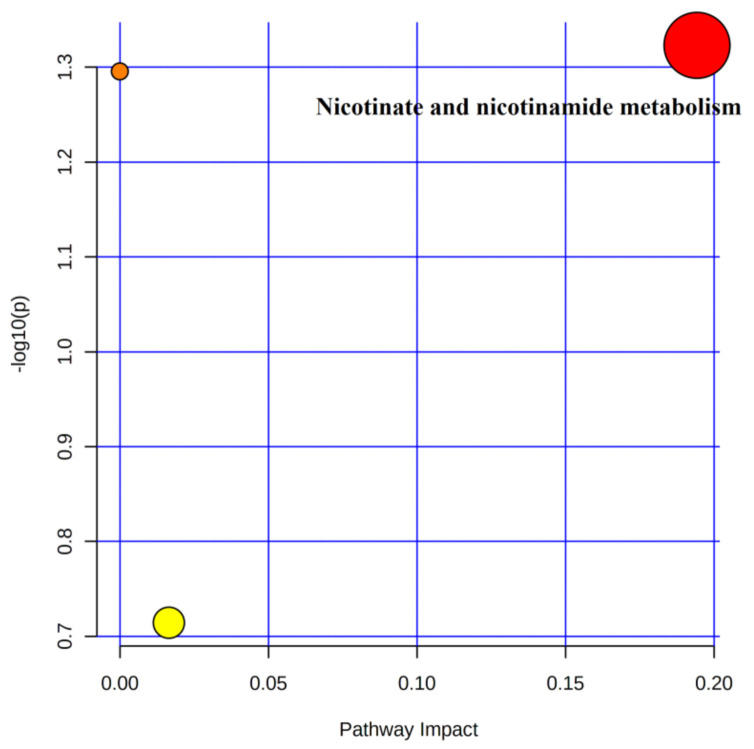
Metabolic pathways of post-training urine and post-competition differential metabolites.

**Table 1 healthcare-09-00381-t001:** Baseline data of athletes ($\bar{X}$ ± SD).

Sex	Number	Age (Years)	Height (cm)	Weight (kg)	BMI	Athlete Level
Female	12	16.46 ± 1.56	172.69 ± 5.14	65.23 ± 7.97	21.84 ± 2.30	Level 1

**Table 2 healthcare-09-00381-t002:** Model validation parameters between groups.

NO.	R2X (cum)	R2Y (cum)	Q2 (cum)	Model Source	Validation Results
1	0.218	0.95	0.633	T3 vs. T1	Established
2	0.287	0.987	0.752	T3 vs. T2	Established

**Table 3 healthcare-09-00381-t003:** Information and trends of differential urinary metabolites at different intervals after water polo games.

No.	Name	Ion	m/z	rt(s)	FC	Trend T2 vs. T1 (*p*)	FC	Trend T3 vs. T2 (*p*)
1	N1-methyl-2-pyridone-5-carboxamide (2-pyridone)	(M + H) +	153.065	85.348	2.61	↑ (0.000042)		
2	Indoleacetic acid	(M + H) +	176.070	224.632	2.47	↑ (0.0045)		
3	Adenosine monophosphate (AMP)	(M + H − H_2_O) +	330.058	289.773	2.091	↑ (0.00046)		
4	N-acetylglutamine	(M + H) +	189.086	313.151	0.44	↓ (0.00412)	0.38	↓ (0.000114)
5	Guanidinosuccinic acid	(M + H) +	176.066	403.111	2.10	↑ (0.000969)		
6	Decanoyl-L-carnitine	M+	316.247	189.977	5.01	↑ (0.00076)	2.13	↑ (0.02635)
7	Trimethylamine N-oxide (TMAO)	(M + H) +	76.075	328.576	0.36	↓ (0.00172)	0.38	↓ (0.0000139)
8	N6-methyl-L-lysine	(M + H − H_2_O) +	143.116	369.352	3.08	↑ (0.0091)		
9	L-Glutamine (L-Gln)	(M + H) +	147.075	225.261	2.41	↑ (0.00288)		
10	1-Deoxynojirimycin	(M + CH_3_COO + 2H) +	224.112	261.669	1.92	↑ (0.0241)		
11	Allopurinol riboside	(M + H − H_2_O) +	251.077	131.79	1.85	↑ (0.0370)		
12	Hypoxanthine (Hx)	(M + H) +	137.044	201.653	6.73	↑ (0.0035)	3.47	↑ (0.0134)
13	Taurine	(M + H) +	126.021	299.017	3.20	↑ (0.00096)		
14	Indole-3-pyruvic acid	(M + NH_4_) +	221.090	289.189	1.85	↑ (0.00267)		
15	Nicotinamide (NAM)	(M + H) +	123.054	64.824	0.49	↓ (0.00005)	0.36	↓ (0.0000062)
16	N-Acetyl-L-Histidine	(M + H) +	198.086	315.946	0.50	↓ (0.023)	0.35	↓ (0.0000089)
17	N6-methyladenosine	(M + H) +	181.071	65.181	2.44	↑ (0.00338)	2.30	↑ (0.00480)
18	Betaine (Bet)	(M + H) +	189.086	287.639	1.49	↑ (0.025)		
19	Palmitoyl ethanolamide	(M + H) +	300.289	36.74	0.72	↓ (0.018)		
20	Primidone	(M + H) +	219.111	225.554	1.92	↑ (0.046)		
21	Dopamine	(M + H − H_2_O) +	136.074	225.573	1.98	↑ (0.0532)		
22	L-methylnicotinamide	M+	137.069	260.072	0.56	↓ (0.054)		
23	Kynurenic acid	(M + H) +	190.048	197.539	2.24	↑ (0.00071)		
24	N-acetylmannosamine	(M + H − H_2_O) +	204.085	257.449	1.72	↑ (0.00828)		
25	4-imidazoleacetic acid	(M + H) +	127.049	341.473			0.32	↓ (0.000028)
26	S-Methyl-5′-thioadenosine	(M + H) +	298.09	99.834			0.70	↓ (0.01918)
27	Phosphorylcholine	(M + H) +	184.072	487.868			0.62	↓ (0.02277)

Note: ↑ indicates upward adjustment, and ↓ indicates downward adjustment.

**Table 4 healthcare-09-00381-t004:** MetPA analysis of metabolic pathways for pre-training and post-competition differential metabolites.

NO.	Pathway Name	Match Statue	*p*	−log(p)	Holm p	FDR	Impact
1	Nicotinate and nicotinamide metabolism	3/15	0.00075822	3.1202	0.06369	0.06369	0.33246
2	Purine metabolism	3/65	0.048181	1.3171	1.0	1.0	0.08068
3	D-Glutamine and D-glutamate metabolism	1/6	0.075082	1.1245	1.0	1.0	0.0
4	Nitrogen metabolism	1/6	0.075082	1.1245	1.0	1.0	0.0
5	Phosphonate and phosphinate metabolism	1/6	0.075082	1.1245	1.0	1.0	0.5
6	Taurine and hypotaurine metabolism	1/8	0.098896	1.0048	1.0	1.0	0.42857
7	Arginine biosynthesis	1/14	0.16689	0.77756	1.0	1.0	0.0
8	Alanine, aspartate and glutamate metabolism	1/28	0.3071	0.51272	1.0	1.0	0.11378
9	Glyoxylate and dicarboxylate metabolism	1/32	0.34284	0.46491	1.0	1.0	0.0
10	Glycine, serine and threonine metabolism	1/33	0.3515	0.45407	1.0	1.0	0.05034
11	Amino sugar and nucleotide sugar metabolism	1/37	0.38506	0.41447	1.0	1.0	0.04723
12	Pyrimidine metabolism	1/39	0.40122	0.39662	1.0	1.0	0.0
13	Tryptophan metabolism	1/41	0.41697	0.3799	1.0	1.0	0.0
14	Tyrosine metabolism	1/42	0.4247	0.37192	1.0	1.0	0.12972
15	Primary bile acid biosynthesis	1/46	0.45464	0.34233	1.0	1.0	0.00758
16	Aminoacyl-tRNA biosynthesis	1/48	0.46906	0.32878	1.0	1.0	0.0

Note: p: original *p*-value from pathway analysis; FDR: error initiation rate; Impact: pathway impact value from topology analysis.

**Table 5 healthcare-09-00381-t005:** MetPA analysis of metabolic pathways for post-training and post-competition differential metabolites.

No.	Pathway Name	Match Statue	P	−log(p)	Holm p	FDR	Impact
1	Nicotinate and nicotinamide metabolism	1/15	0.04752	1.3231	1.0	1.0	0.1943
2	Histidine metabolism	1/16	0.050622	1.2957	1.0	1.0	0
3	Purine metabolism	1/65	0.19304	0.714435	1.0	1.0	0.01651

Note: p: original *p*-value from pathway analysis; FDR: error initiation rate; Impact: pathway impact value from topology analysis.

## Data Availability

The data presented in this study are available on request from the first author.

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
