# Peer review of "Mechanism of the Effect of High-Intensity Training on Urinary Metabolism in Female Water Polo Players Based on UHPLC-MS Non-Targeted Metabolomics Technique"

_healthcare, 2021, doi:10.3390/healthcare9040381_

Round 1
Reviewer 1 Report
Based on LC-MS metabolomic assays and multiple analysis methods, 27 differential metabolites and two metabolic pathways were screened after water polo matches compared to before and after training. The differential changes in urine after matches were related to the 448 changes of the common metabolic pathway before and after training: niacin and niacinamide metabolism.
I found the work interesting and well prepared. But I also feel that discussion with other sports results could be ameliorated. There is space for discussion with other sports. For instance it is not clear, and in order to reduce experimental variability, all athletes were trained at the Shanxi Sports Center throughout the experiment, and their team teachers were responsible for their diet and living. Was the water treated with chlorine or salt/open or indoor? The synthesis of chloramines, which result from the combination chlorine disinfectants and perspiration, oils and urine that enter pools on the bodies of swimmers, could have influence in the metabolic profile and lung function. I believe that authors could easily highlight the relevance of the type of water in their discussion.
Reviewer 2 Report
The manuscript of Wang et al. is quite interesting. The Authors describe the changes in the concentration of metabolic-derived compounds observed in urine samples collected from athletes competing in water polo competitions.
Tipographical errors:
Pag 3, line 92: not “ml” but “mL”
Pag 4, line 134: not “to. mzXML” but “to mzXML”
Legend of Table 4:: not “metabolitesdifferential” but “metabolites differential”
There are only a few issues to clarify.
- It would be necessary to clarify whether the experimental conditions of the HPLC analyses have been optimized or taken from the literature (insert reference).
- The training is described in detail, but the Authors should insert a sentence to clarify whether this training is standard or planned for the study.
- Line 66: "their team teachers were responsible for diet of athlete." It would be advisable to specify the diet observed by the athletes. Indicate whether the diet is high in carbohydrates or proteins and whether a diet common to the athletes was maintained or planned for the study. In fact, the Authors state that the type of diet is important because it affects the metabolic pathways and therefore the metabolites present in the urine.
- In Discussion, first line: “Urine is ideal for metabolimics studies because of its sterility”. Clarify the concept. Urine is not always sterile. Have the urine samples been subjected to microbiological analysis?
- Line 440: “therefore, this study recommends that athlete meals be adjusted to a reasonable carbohydrate to protein ratio”. This claim is already proven by numerous studies. The Authors do not propose an appropriate report. It would be appropriate to include some references from previous studies that are consistent with this statement.
Provided that these issues are settled the paper is worthy of pubblication.
